# A Proteomic Atlas of Lineage and Cancer-Polarized Expression Modules in Myeloid Cells Modeling Immunosuppressive Tumor-Infiltrating Subsets

**DOI:** 10.3390/jpm11060542

**Published:** 2021-06-11

**Authors:** Ester Blanco, Maria Ibañez-Vea, Carlos Hernandez, Lylia Drici, Xabier Martínez de Morentin, Maria Gato, Karina Ausin, Ana Bocanegra, Miren Zuazo, Luisa Chocarro, Hugo Arasanz, Gonzalo Fernandez-Hinojal, Joaquin Fernandez-Irigoyen, Cristian Smerdou, Maider Garnica, Miriam Echaide, Leticia Fernandez, Pilar Morente, Pablo Ramos-Castellanos, Diana Llopiz, Enrique Santamaria, Martin R. Larsen, David Escors, Grazyna Kochan

**Affiliations:** 1Oncoimmunology Group, Navarrabiomed, Fundación Miguel Servet-Complejo Hospitalario de Navarra UPNA-IdISNA, 31008 Pamplona, Spain; ester.blanco.palmeiro@navarra.es (E.B.); maria.ibanez@unavarra.es (M.I.-V.); chernans77@gmail.com (C.H.); mariagato13@gmail.com (M.G.); ai.bocanegra.gondan@navarra.es (A.B.); mtxu26@gmail.com (M.Z.); luisa.chocarro.deerauso@navarra.es (L.C.); hugo.arasanz.esteban@navarra.es (H.A.); gfhinojal@gmail.com (G.F.-H.); mgarnics@navarra.es (M.G.); mechaidg@navarra.es (M.E.); leticia.fernandez.rubio@navarra.es (L.F.); pilar.morente.sancho@navarra.es (P.M.); pramosca@navarra.es (P.R.-C.); descorsm@navarra.es (D.E.); 2Genetics, Genomics and Microbiology Research Group, Institute for Multidisciplinary Research in Applied Biology (IMAB-UPNA), University Public of Navarre, Campus Arrosadía, 31006 Pamplona, Spain; 3The Novo Nordisk Foundation Center for Protein Research, Faculty of Health and Medical Sciences, University of Copenhagen, Blegdamsvej 3B, 2200 Copenhagen, Denmark; lylia.drici@cpr.ku.dk; 4Bioinformatics Group, Navarrabiomed Biomedical Research Center, Institute of Health Carlos III (ISCIII), Navarra Institute for Health Research (IdiSNA), Irunlarrea 3, 31008 Pamplona, Spain; xmartind@navarra.es; 5Proteored-ISCIII, Proteomics Platform, Navarrabiomed, Complejo Hospitalario de Navarra (CHN), Universidad Pública de Navarra (UPNA), IdISNA, Irunlarrea 3, 31008 Pamplona, Spain; kausinpe@navarra.es (K.A.); jfernani@navarra.es (J.F.-I.); esantamma@navarra.es (E.S.); 6Division of Gene Therapy and Regulation of Gene Expression, Cima Universidad de Navarra and Instituto de Investigación Sanitaria de Navarra (IdISNA), 31008 Pamplona, Spain; csmerdou@unav.es; 7Centro de Investigación Médica Aplicada (CIMA), Universidad de Navarra, 31008 Pamplona, Spain; diallo@unav.es; 8Centro de Investigación Biomédica en Red de Enfermedades Hepáticas y Digestivas (CIBEREHD), 31008 Pamplona, Spain; 9IdiSNA, Instituto de Investigación Sanitaria de Navarra, 31008 Pamplona, Spain; 10Department of Biochemistry and Molecular Biology, University of Southern Denmark, Campusvej 55, 5230 Odense M, Denmark; mrl@bmb.sdu.dk

**Keywords:** myeloid-derived suppressor cells, cancer, tumor-infiltrating macrophages

## Abstract

Monocytic and granulocytic myeloid-derived suppressor cells together with tumor-infiltrating macrophages constitute the main tumor-infiltrating immunosuppressive myeloid populations. Due to the phenotypic resemblance to conventional myeloid cells, their identification and purification from within the tumors is technically difficult and makes their study a challenge. We differentiated myeloid cells modeling the three main tumor-infiltrating types together with uncommitted macrophages, using ex vivo differentiation methods resembling the tumor microenvironment. The phenotype and proteome of these cells was compared to identify linage-dependent relationships and cancer-specific interactome expression modules. The relationships between monocytic MDSCs and TAMs, monocytic MDSCs and granulocytic MDSCs, and hierarchical relationships of expression networks and transcription factors due to lineage and cancer polarization were mapped. Highly purified immunosuppressive myeloid cell populations that model tumor-infiltrating counterparts were systematically analyzed by quantitative proteomics. Full functional interactome maps have been generated to characterize at high resolution the relationships between the three main myeloid tumor-infiltrating cell types. Our data highlights the biological processes related to each cell type, and uncover novel shared and differential molecular targets. Moreover, the high numbers and fidelity of ex vivo-generated subsets to their natural tumor-shaped counterparts enable their use for validation of new treatments in high-throughput experiments.

## 1. Introduction

The tumor microenvironment (TME) is composed of cancer cells and other cell types which include tumor-infiltrating immune cells. Cancer cells re-program immune cells to inhibit their anti-tumor activities through production of a range of immunosuppressive molecules [1]. Tumor-infiltrating myeloid cells constitute an important component of the TME, where they promote tumor growth, metastasis and enhance resistance to anti-cancer treatments. Among these, myeloid derived suppressor cells (MDSCs) and tumor associated macrophages (TAMs) are major tumor-promoting cells within the TME [2,3]. MDSCs and TAMs exert strong T cell-inhibitory functions through cell-to-cell contacts and production of immunosuppressive cytokines [4,5,6]. In advanced cancers, their expansion contributes to broad systemic immunosuppression in patients [7]. In this way, TAMs and MDSCs overtake the control of anti-tumor immune responses in the patient. Their increase is a clinical marker of poor prognosis, and therefore they constitute potential targets for improving anti-cancer therapies [8].

MDSCs are considered a heterogeneous group of immature myeloid cells, arising from pathological myelopoiesis particularly in cancer [8,9]. In mice, MDSCs are divided in 2 distinct phenotypes, monocytic (m-MDSC, CD11b^+^ Ly6C^high^ Ly6G^neg^) and granulocytic MDSC (g-MDSC, CD11b^+^ Ly6C^low/neg^ Ly6G^high^), which resemble either monocytes or neutrophils, respectively [5]. Humans also have equivalent populations albeit expressing different markers from those of mice [2]. Due to the phenotypic resemblance to conventional myeloid cells, their identification and purification from within the tumors is a technical challenge. Indeed, many authors consider m-MDSCs and g-MDSCs as pathologically activated monocytes or neutrophils, respectively, rather than cell lineages of their own. A recent study employed a range of high-throughput techniques to delineate the differences between neutrophils and g-MDSCs [6]. The ontogeny and differentiation relationships between m-MDSCs and g-MDSCs are also difficult to establish. On one hand, m-MDSCs may be related to monocytes and g-MDSCs to neutrophils. On the other hand, others and we previously showed that g-MDSCs could be differentiated from m-MDSCs both in in vitro and in vivo murine models [5,10], making monocytic MDSCs direct precursors of granulocytic MDSCs.

TAMs also constitute a classical major population of immune cells within the TME associated to cancer progression and poor prognosis, which share characteristics with M2 macrophages [11] and with monocytic MDSCs. Indeed, TAMs and m-MDSCs share many features and they may also have common monocytic precursors [12]. In agreement with this, it has been shown that m-MDSCs can differentiate towards TAMs [13]. While there are many studies on macrophages or MDSC, their closely related functions, phenotypic similarities and differentiation plasticity contributes to the confusion over their ontogeny and differential characteristics.

In the present study, we generated a detailed atlas with proteomic signatures between myeloid cells modeling monocytic, granulocytic MDSCs and TAMs related to lineage, and cancer polarization. These analyses shed light on the similarities and differences between the myeloid immunosuppressive populations, their potential mechanisms of immune regulation, and highlighted differences identifying each cell type with unique proteomic fingerprints.

## 2. Materials and Methods

### 2.1. Cells and Mice

Approval for animal studies was obtained from the Animal Ethics Committee of the University of Navarra and of the Government of Navarra (Reference: 077-19). Bone marrow cells were obtained from the femur and tibia of C57BL/6 mice and ex-vivo cultured in either RPMI or DMEM as described before [14]. B16F0 cells constitutively expressing mouse GM-CSF (B16F0-GMCSF) were generated and described before [5,15]. Briefly, cells were transduced with lentivectors encoding GM-CSF followed by selection by puromycin resistance and cell cloning by limiting dilution. In a similar way, B16F0-MCSF cells were generated by transduction with lentivectors encoding a soluble version of murine macrophage colony stimulating factor (MCSF) in a pDUAL-PuroR backbone described before [5,15].

### 2.2. In Vitro Differentiation and Purification of Myeloid-Derived Suppressor Cells, Tumor-Associated Macrophages and Non-Polarized Macrophages

M2-like TAMs and MDSCs were obtained following a standard protocol described in [5,15]. Briefly, bone marrow cells were cultured in conditioned medium from B16F0-GMCSF or B16F0-MCSF for 6 days as described previously [5,15]. As controls, uncommitted macrophages (M0) were differentiated from bone marrow cells by incubation for 6 days in complete RPMI medium containing 10 ng/mL recombinant CSF-1/M-CSF, and supplemented with 10% FBS, penicillin/streptomycin (Sigma-Aldrich; St Louis, MO, USA) and 2 mM L-glutamine (Sigma-Aldrich; St Louis, MO, USA). The morphology of myeloid populations was evaluated by microscopy using Cytation 5 (Biotek; Winooski, VT, USA).

Granulocytic and monocytic MDSCs were purified using the myeloid-derived suppressor cell isolation kit (Miltenyi Biotec; Bergisch, Germany) according to the manufacturer’s recommendations. Purity was confirmed by flow cytometry with the appropriate markers for each population.

### 2.3. Cell Staining and Flow Cytometry

Surface and intracellular staining were performed as described previously [16,17] using the indicated antibodies: From Biolegend (San Diego, California, USA): Brilliant Violet 510™ anti-mouse I-A/I-E antibody, PE/Cyanine7 anti-mouse Ly-6G antibody and APC anti-mouse CD80. From BD Bioscience: FITC Alfa-Antimouse IgG. From Milteny (Bergisch, Germany): APC-Vio^®^ 770 anti-mouse Ly-6C, PE REAfinity™ anti-mouse F4/80, Ly-6C, Biotin, FITC anti-mouse and PE anti-mouse IgG1 antibody. From Invitrogen (Waltham, MA, USA): Alexa Fluor 488 anti-rabbit IgG (H+L) highly cross-adsorbed secondary antibody and APC anti-mouse CD11c. From Tonbo: PerCP-Cyanine5.5 Anti-Human/Mouse CD11b (M1/70).

Data were collected using the FACSCanto Flow Cytometer (BD Biosciences, Franklin Lakes, NJ, USA) and analyzed with Flowjo.

### 2.4. Western Blotting

Immunoblots were carried out as described before [14] with the following antibodies: Mouse anti- STAT3 antibodies and polyclonal anti-phosphorylated STAT3 molecules from Cell Signaling Technology (Danvers, MA, USA); anti-beta-actin (SIGMA; St. Louis, MO, USA). From BD Bioscience (Franklin Lakes, NJ, USA), mouse anti-pan ERK. From Cell Signaling (Danvers, MA, USA), rabbit anti-mouse HIF-1α (D2U3T) (#14179). Peroxidase-conjugated polyclonal anti-mouse and anti-rabbit antibodies were purchased from DAKO.

### 2.5. Mass Spectrometry-Based Quantitative (Shotgun) Proteomics and Bioinformatics Analyses

Three independent proteomic experiments were carried out. The first two experiments used three biological replicates per sample (MDSC, TAM and M0 for the first experiment; m-MDSC and TAM for the second experiment), and two biological replicates for the third experiment (m-MDSC and g-MDSC).

First and second experiments: Cell pellets were homogenized in a lysis buffer containing 7 M urea, 2 M thiourea 50 mM DTT. The homogenates were spun down at 14,000× *g* for 1 h at 15 °C. Before proteomic analysis, protein extracts were precipitated with methanol/chloroform and pellets dissolved in 6 M urea and Tris 100 mM pH 7.8. Protein quantitation was performed with the Bradford assay kit (Bio-Rad, Hercules, CA, USA). Protein enzymatic cleavage was carried out with trypsin (Promega; Madison, WI, USA; 1:20, *w*/*w*) at 37 °C for 16 h as previously described [18]. Peptide mixtures were separated by a reversed-phase chromatography using an Eksigent NanoLC ultra 2D pump fitted with a 75 mm ID and 25 cm length column (Thermo Scientific, Waltham, MA, USA). Samples were first loaded for desalting and concentration into a 0.5 cm length 100 mm ID precolumn packed with the same chemistry as the separating column. Mobile phases were 100% water and 0.1% formic acid (Buffer A) and 100% acetonitrile and 0.1% formic acid (Buffer B). Column gradient was developed in a 240-min two-step gradient from 5% B to 25% B in 210 min and 25% B to 40% B in 30 min. Eluting peptides were analyzed using a 5600 Triple-TOF system, as previously described [19]. MS/MS data acquisition, searching, peptide quantitation, and statistical analysis were performed as previously described [19]. Progenesis software was used to obtain quantitative data. Briefly, raw data were submitted to mathematical algorithms to remove background, to align and compensate the “between-run” variation, and to choose the same peaks in all samples in the peak-picking phase. Then, peptides were identified with the information obtained using Protein Pilot software. Output files with the identified proteins were then managed with Perseus [20] for subsequent statistical analysis. An unpaired Student t-test was used for direct comparisons between two groups of samples. Statistical significance was set at *p* < 0.05 in all cases and 1% peptide false discovery rate threshold was considered (calculated based on the search results against a decoy database). In addition, an absolute fold change of <0.77 (downregulation) or >1.3 (upregulation) in linear scale was considered to be significantly differentially expressed.

Third experiment: sample preparation; proteins were extracted and digested according to the method used in [21]. Briefly, cells were lysed with sodium bicarbonate, proteases and phosphatase inhibitors. Membrane proteins were isolated by ultracentrifugation. The supernatant containing the soluble fraction was digested on filter (Sartorius Vivacon, Gottingen, Germany (500 µL; 10 KDa) centrifugal Filters). The pellet containing membrane proteins was redissolved in 40 µL of 6 M urea/2 M thiourea. The proteins were reduced (10 mM dithiothreitol (DTT)) and alkylated (20 mM IAA) prior digestion. Then they were digested with Lys-C (1:100) for 3 h, and with trypsin (1:50) overnight at room temperature after diluting to a final urea concentration of 1 M for membrane proteins. Then, the solutions were acidified with formic acid (2%) to stop the digestion and precipitate lipids. The samples were desalted and stored to the subsequent labeling step. An aliquot of the elution was taken to measure peptide concentration by Qubit. Equal amounts of peptides from each condition were TMT 6-plexTM labeled according to manufacturer’s guidelines (Thermo Fisher Scientific). Labeled peptides were combined in 1:1:1:1:1:1 proportion based on the Qubit quantification and nLC-MS/MS analysis. The conditions were as follows: m-MDSC_Rep 1: TMT 126; g-MDSC_Rep 1: TMT 127; m-MDSC_Rep 2: TMT 128; g-MDSC_Rep 2: TMT 129; m-MDSC_Rep 3: TMT 130; g-MDSC_Rep 3: TMT 131.

Sample fractionation: samples from the third experiment were fractionated by high pH reversed-phase (RP) and HILIC. Samples were fractionated by HILIC [22] using an Agilent Technologies 1200 rapid resolution liquid chromatographic system (Agilent, Santa Clara, CA, USA). Separation was achieved on a packed TSK Gel Amide 80 (3 μm; Tosoh Bioscience, Stuttgart, Germany) (15 cm × 0.32 mm) micro-capillary column. Mobile phase was based on water (A) and 90% ACN (B), acidified with 0.1% TFA. Forty microliters of sample was loaded on the column at a flow rate of 12 µL/min for 8.6 min and then peptides were separated with a gradient from 100 to 60% B for 26.4 min at a flow rate of 6 µL/min. Fractions were collected at different times through the chromatogram and combined into 15 fractions according to the UV chromatogram.

High pH RP fractionation: a Dionex liquid chromatographic system was used to fractionate the samples by high pH RP chromatography. Peptides were separated in an Acquity UPLC^®^ M-Class CSHTM C18 (130 Å; 1.7 µm (300 µm × 100 mm)) column at a flow rate of 5 µL/min. The mobile phase was composed of 20 mM ammonium formate (pH = 9.2–9.3) (A) and 80% ACN in A (B). Peptides were separated with a three-step gradient from 2 to 40% B for 78 min, from 40 to 50% B for 32 min and from 50 to 95% B for 5 min. Fractions were collected every 166 s, concatenated 3 times and combined into a total of 15 fractions.

Analysis of the samples by nano-LC-MS/MS: samples were resuspended in 6 µL of 0.1% formic acid to their analysis. Peptides were loaded on a custom-made ReproSil-Pur 120 AQC18 (Dr. Maisch GmbH, Ammerbuch, Germany) pre-column (2 cm × 100 µm, 5 µm) and separated on a ReproSil-Pur 120 AQC18 (Dr. Maisch GmbH; Ammerbuch, Germany) column (20 cm × 75 µm, 3 µm) using a nano EasyLC system (Thermo Scientific, Waltham, MA, USA) and eluted at a flow of 250 nL/min. Mobile phase was 95% acetonitrile (B) and water (A) both containing 0.1% formic acid. Peptides were separated using a different gradient depending on the samples.

Mass spectrometric measurements were performed in a Q Exactive HF MS system (Thermo Scientific, Waltham, MA, USA). MS scans (400–1600 m/z) were acquired in the orbitrap at a resolution of 120,000 (at 200 m/z) with a maximum injection time (IT) of 120 ms and AGC target of 3·106. Data-dependent MS/MS analysis for the 12 most intense ions (minimum AGC target of 2·103) isolated in the quadrupole with an isolation window of 1.2 m/z were performed in the orbitrap at a resolution of 30,000 using HCD fragmentation (NCE 34%) with a maximum injection time of 200 ms and an AGC target of 1·105. Dynamic exclusion was activated for 8 s. Data was acquired using Thermo Xcalibur version 3.0.63 software.

### 2.6. Proteomic Data Analysis

The raw data were processed and quantified by Proteome Discoverer (version 2.1, Thermo Scientific, Waltham, MA, USA) against Uniprot mus musculus reference database by using Mascot (v2.3, Matrix Science Ltd., London, UK). Database searches were performed using the following parameters: precursor mass tolerance of 10 ppm, product ion mass tolerance of 0.05 Da, 1 missed cleavages for trypsin, TMT 6-plex labelling on protein N-terminal and Lys as fixed modifications, and carbamidomethylation, as dynamic modifications. The TMT datasets were quantified using the centroid peak intensity with the “reporter ions quantifier” node. Only peptides with up to a q-value of 0.01 (Percolator), Mascot Rank 1, maximum ΔCn of 0.05, and a cut off value of Mascot score ≥22 were considered for further analysis.

Data normalization and significance analysis: Two biological replicates were considered for the significance analysis in the third experiment. The log2 values of the measured signal-to-noise (third experiment) values were normalized with the median. Quantification of proteins was obtained by merging the peptides with the R Rollup function (http://www.omics.pnl.gov; accessed on 5 May 2021) considering at least two unique peptides not allowing for one-hit-wonders and using the mean of the normalized values for each peptide. Then the mean over the experimental conditions and replicates for each peptide was subtracted in order to decrease the influence of measurement errors. In the case of the third experiment, only proteins showing a ratio higher or lower to a two standard deviation were considered up- or down-regulated.

Bioinformatic analyses: Construction of functional interactomes from up- or down-regulated proteins was carried out with STRING (Search Tool for the Retrieval of Interacting Genes) analysis tool (V.9.1) using high confidence interaction score (>0.7) and with the Ingenuity Pathway Analysis Tool (Qiagen, Hilden, Germany) (https://www.qiagen.com/us/products/discovery-and-translational-research/next-generation-sequencing/informatics-and-data/interpretation-content-databases/ingenuity-pathway-analysis/; accessed on 5 May 2021). The specific protein sets and comparisons were selected as indicated in results. Tfacts algorithm (https://www.tfacts.org/TFactS-new/TFactS-v2/index1.html; accessed on 5 May 2021) was used to identify potentially activated/de-activated transcription factors using the indicated differential proteomes.

Data repository: MS data and search results files were deposited in the Proteome Xchange Consortium via the JPOST partner with the identifier PXD025708 for ProteomeXchange and JPST001146 for jPOST (for reviewers: https://repository.jpostdb.org/preview/143364250608a8e370fba3; accessed on 5 May 2021; Access key: 4446).

### 2.7. Statistical Analyses

GraphPad Prism software package was used for plotting data and statistical analyses. No data was considered an outlier. Normality of variables was tested with Kolmogorov–Smirnov tests. Data groups exhibiting a normal distribution were analysed by t-tests (two independent group experiments), or one-way ANOVA and Tukey’s pair-wise comparisons for multi-comparison studies. In previous studies, we confirmed that mean fluorescence intensities from surface and intracellular flow cytometry stainings were normally distributed in our cell samples [4,5,16,17,23,24]. Therefore, in these cases, parametric tests were used using either one-way or two-way ANOVAS.

## 3. Results

### 3.1. Ex Vivo Differentiation of Myeloid Cells Modeling Regulatory Subpopulations

To generate myeloid cell preparations modeling tumor-infiltrating subsets, we applied an ex vivo-differentiation system that mimics the conditions within the TME. This system was previously developed to generate MDSCs that closely resemble endogenous tumor-infiltrating MDSCs, and has been validated in several murine cancer models [5,15,23,25,26,27,28,29]. The protocol was here adapted as well for differentiation of macrophages resembling TAMs (Materials and Methods). As controls, non-polarized macrophages (M0) were differentiated following standard conditions.

To confirm the identity of the cell subsets differentiated ex vivo, cell culture preparations were examined by phase contrast (Figure 1a) and immunocytochemistry for the detection of the macrophage-specific marker F4/80 (Figure 1b). Both M0 and TAM macrophages exhibited the prototypical elongated shapes and high F4/80 expression. TAM cells appeared as longer spindle-like compared to non-polarized macrophages. As expected, F4/80 expression was virtually absent in MDSCs, which also showed prototypical mulilobed and annular nuclei.

To further confirm their identity, the expression of MHC-II and CD80 was evaluated by flow cytometry (Figure 1c). In mice, macrophages constitutively express high levels of surface MHC II, which was confirmed in ex vivo differentiated M0 and TAM. In contrast, MHC II was strongly down-modulated in MDSCs, in agreement with previous studies [5,15,27].

CD80 is a classical activation marker in myeloid cells, which is broadly down-regulated in tumor-infiltrating subsets. Accordingly, CD80 expression was abrogated in TAMs and reduced in MDSCs compared to control uncommitted M0 macrophages. These results confirmed that these ex-vivo differentiated myeloid cells resembled tumor-associated subsets.

### 3.2. Proteome Profiles of Ex Vivo-Differentiated Myeloid Subsets Separate Lineage-Regulated from Tumor-Polarized Interactome Modules

To gain insight into lineage and tumor-induced differences between the ex vivo-differentiated myeloid cell subsets, we performed label-free quantitative proteomics, using three independent biological replicates per myeloid culture. A total of 1564 proteins were identified with a FDR lower than 1%. Proteins with significant changes between the groups (*p* < 0.01) were selected for further studies. A principal component analysis confirmed that ex vivo-differentiated myeloid cell cultures were homogeneous according to type (Appendix A). An unbiased cluster analysis of significantly regulated proteins within the three proteomes grouped M0 and TAM macrophages, according to global similarities of their proteomes (Figure 2a). These results confirmed that myeloid cells were broadly grouped by lineage, supporting the validity of the differentiation methods and the proteomic data.

In contrast to M0 macrophages, ex vivo-differentiated TAM and MDSCs were exposed to cancer cell-derived conditioning media. We used this fact to uncover shared regulated protein modules arising from reprogramming by cancer cells. To achieve this, proteins commonly-regulated by TAM and MDSCs were identified, clustered and compared to uncommitted M0 macrophages (Figure 2b,c). Cancer-driven reprogramming caused among other processes the up-regulation in both TAM and MDSCs of metabolic pathways, drug metabolism, metabolism of nitrogen compounds and regulation of immune responses (Figure 2d). Likewise, other functional pathways were down-modulated including positive regulation of immune responses, vesicle mediated transport and some metabolic processes (Figure 2e). These proteome modules were used to infer the transcription factors regulating cancer-polarization using Tfacts. MYC and SP1 were predicted to be major regulators, in agreement with published experimental data [30,31] (Appendix A), which further validated our ex vivo differentiation systems.

Then, we wondered whether we could identify cancer-regulated pathways that were specific for either TAMs or MDSCs. Therefore, the proteome of M0 uncommitted macrophages was used as a normalizing control between the proteomes of TAM and MDSC preparations, uncovering 466 and 751 differentially expressed proteins in TAMs and MDSCs, respectively (Figure 2c). MDSC preparations in these experiments consisted of 40% monocytic and 60% granulocytic subsets. These proteins were then separated into overexpressed or down-modulated proteins. Then, the differential functional interactomes up-regulated by TAMs or MDSCs were constructed with STRING. TAMs up-regulated processes including lipid metabolism, phagosome and immune response (Figure 3a), while MDSCs preferentially increased drug metabolism and response to stress and nitrogen compound metabolism (Figure 3b).

### 3.3. Kinase Profiles Associated to Cancer-Polarized Immunosuppressive Myeloid Cells

To identify kinase networks regulating the cancer-driven interactome modules, we identified the kinases present in the proteomes. CDK1 and STK26 were found to be the most upregulated in tumor-polarized myeloid cells (Figure 4a). Upstream regulators were further identified with Ingenuity Pathway Analysis, highlighting MAPK1 (Figure 4b). STAT3 was found downstream participating in the signaling route, which was corroborated by western blot (Figure 4c). STAT3 functions are regulated through differential phosphorylation in two residues (Y705, S727). Interestingly, there were differences in STAT3 phosphorylation between TAMs and MDSCs. While STAT3 was phosphorylated in S727 both in TAM and MDSCs, STAT3 was preferentially Y705-phosphorylated only in TAMs.

### 3.4. Unique Protein Expression Profiles Discriminate Ex Vivo-Differentiated Monocytic MDSCs from TAMs

Some studies have suggested that monocytic MDSCs are closely related to TAMs and not so much to g-MDSCs, because TAMs and m-MDSCs share many phenotypic and functional features. To uncover the similarities and differences between m-MDSCs and TAMs, monocytic MDSCs were purified to homogeneity from ex vivo-differentiated MDSCs and their proteome compared to TAMs. A total of 1536 proteins were uniquely identified. Of these, 336 were differentially regulated supporting the evidence for these myeloid populations are distinct from each other (Figure 5a). A total of 188 proteins were significantly increased in m-MDSCs, while 148 proteins were highly expressed in TAMs. Functional proteomic interactome networks were constructed with STRING and classified by gene ontology and KEGG pathways. TAMs upregulated functions related to lysosomes and phagosomes with a proteome indicative of an upregulated cholesterol metabolism and increased endocytosis (Figure 5b). In contrast, m-MDSCs activated pathways regulating leukocyte transendothelial migration, reorganization of actin cytoskeleton, upregulation of pentose phosphate pathway, and increased glycolysis and gluconeogenesis (Figure 5c). Significant upregulation of proteins controlling Fc-gamma receptor-mediated phagocytosis and activation of VEGFA-VEGFR2 signaling pathways were also detected in m-MDSCs but not in TAMs. Finally, m-MDSCs upregulated ROS detoxification mechanisms, in agreement with previous results [15]. The differential proteome in m-MDSCs and TAMs exhibited different regulating networks of immune responses. TAMs up-regulated MHCII-dependent antigen presentation, toll like receptor cascades and L1CAM (Figure 5d). In contrast, m-MDSCs upregulated IL-17, VEGFA-VEGFR2 and chemokine signaling pathways (Figure 5e). Monocytic-MDSCs showed functional interactome networks associated to ROS and RNS in phagosomes, and the expression of matrix metaloproteases MMP8 and MMP9, absent in TAMs. Interestingly, both subsets differed in production of specific components of the complement cascade. TAMs upregulated C1q subunit b (FC > 52.61), but on the other hand m-MDSCs significantly increased C3 expression (FC > 18.9).

Kinases were selected from the proteomes of TAMs and monocytic MDSCs and their expression compared to identify profiles separating both populations. There were significant differences in kinase profiles between the two subsets (Figure 5f). MAPK3 was up-regulated in m-MDSCs compared to TAMs. We then evaluated the expression ERK1 and ERK2 by western blot, two downstream kinases of MAPK3 associated to immunosuppression. Indeed, ERK1 and ERK2 were strongly up-regulated in m-MDSCs but not in TAM. As we found differences in STAT3 phosphorylation between MDSC, TAM and M0 subsets, the status of STAT3 phosphorylation was also evaluated by western blot in monocytic MDSCs and TAMs, confirming the differential Y705 STAT3 phosphorylation in TAMs (Figure 5g). In this case, we used S727-phosphorylated STAT3 as a loading control, as we found in this study not to be different between our ex vivo-differentiated myeloid cells. We then wondered whether activities of distinct transcription factors could also discriminate monocytic MDSCs from TAMs. Therefore, Tfacts was used to associate activities of transcription factors with the differentially regulated proteome between these two myeloid subsets (Figure 5h). HIF1-alpha was predicted as a principal transcription factor upregulated in m-MDSC compared to TAMs, which was confirmed by western blot (Figure 5h).

### 3.5. Key Nuclear Differential Pathways and Major Metabolic Routes Separate Monocytic from Granulocytic MDSCs

Monocytic-MDSCs phenotypically resemble classical monocytes while granulocytic MDSCs to neutrophils. However, others and we showed that g-MDSCs could be ex vivo differentiated from m-MDSCs [5,10]. To highlight the differences and similarities between the two MDSC subsets, MDSCs were produced and both populations isolated and purified to homogeneity. To increase the sensitivity of our analyses, cytoplasmic and membrane fractions were separated followed by quantitative proteomics and side-by-side comparisons of proteome profiles (Figure 6a,b). A total of 6071 proteins were identified in membrane fractions and 4576 soluble proteins with an FDR lower than 1%. From these, the expression of 220 soluble proteins and 310 membrane proteins was differentially regulated between the two subsets (Figure 6b). Monocytic-MDSCs had significantly increased 203 soluble proteins and 36 membrane proteins, while g-MDSCs upregulated 17 soluble proteins and 274 membrane proteins. Functional protein networks were constructed with STRING (Figure 6c) and the differential pathways identified by gene ontology analyses and KEGG pathways (Figure 7a). Major differences were found on the profiles of proteins involved in immune response modulation and metabolic interactomes, although all of them related to negative regulation of T cell and interferon responses (Figure 7b). Monocytic MDSCs exhibited functions related to antigen processing and presentation. In contrast, granulocytic MDSCs had up-regulated stress-response pathways involving the mitochondrion, FGF signaling and alternative RNA processing/metabolism in the nucleus, changes in DNA conformation as well as upregulation of proteins involved in senescence (Figure 6c and Appendix A). Importantly, alternative splicing was activated together with altered transcription regulation (Figure 6c and Appendix A).

To identify transcription factor profiles that could discriminate between the two MDSC populations, the Tfacts algorithm was used with the g-MDSC differential proteome. Interestingly, transcription factors associated to inflammation, immunosuppression and TGF-beta signaling were predicted to be active in g-MDSC compared to m-MDSC (Figure 7c).

## 4. Discussion

Tumor-infiltrating myeloid subsets are known for their immunosuppressive character and oncogenic promoting activities, especially by enhancing tumor progression. Myeloid-derived suppressor cells, tumor-associated macrophages and tolerogenic dendritic cells constitute the mayor players in tumor progression. As myeloid subsets are highly plastic and share many surface markers, in many instances is difficult to identify the subset of interest as a cellular target for therapies or studies. Numerous markers have been proposed to describe specific subsets and differentiation stages. However, the isolation of tumor-infiltrating myeloid cell types is a technical challenge due to their low abundance, but also to their high degree of phenotypic plasticity. To circumvent this caveat, we previously developed an ex vivo differentiation system of MDSCs that closely resemble tumor-infiltrating counterparts. This system relied on differentiation of MDSCs with conditioned medium from murine melanoma B16 cells modified to constitutively express GM-CSF. These ex vivo-generated MDSCs have been successfully validated in several murine cancer models by us and others, and constitute good models for bona fide intra-tumor MDSCs [5,15,23,25,26,27,28,29]. Here we modified the protocol to differentiate macrophages that resemble TAMs. These ex vivo-generated TAMs showed characteristics identical to those published for M2-polarised TAMs and were phenotypically and morphologically different from control, non-polarized macrophages. Then, by differentiating MDSCs and purifying them to homogeneity into monocytic and granulocytic subsets, we performed side-by-side proteomic analyses between the different subsets to uncover their relationships and differences. The analysis of all data including validation of key targets uncovered a hierarchy of cellular functions (Figure 8a), kinases and transcription factors (Figure 8b) between these populations. Compared to uncommitted macrophages, tumor-polarized myeloid TAMs and MDSCs were characterized by the predicted activities of c-MYC, SPI and STAT3. The status of STAT3 was validated by western blot. Despite their numerous similarities such as the expression of CD206, CD163 and others (not shown) we found that while TAMs and MDSCs shared S727-STAT3 phosphorylation, only TAMs had STAT3 phosphorylated in Y705 (Figure 8b). A more precise comparison between TAMs and purified monocytic MDSCs again demonstrated TAM-specific Y705-STAT3 phosphorylation. STAT3 phosphorylation in Y705 is required for dimerization and transcriptional activation [32]. In agreement with our data, this phosphorylation in STAT3 was shown to polarize macrophages towards an M2 phenotype in models of salmonella infection, myocardial infraction and cancer [33,34,35]. Moreover, our proteomic data indicated that LIPA was up-regulated specifically in TAMs. LIPA is a protein that contributes to macrophage polarization towards M2, and critical for their activation. LIPA participates in lipolysis of fatty acids that have been up-taken by the scavenger receptor CD36 [36], also found upregulated in this study in TAMs. On the other hand, phosphorylation of S727 in STAT3 causes its translocation to the mitochondria, where it interacts with elements of the electron transport. This mechanism regulates mitochondrial oxidative phosphorylation and protects cells from ROS and apoptosis [37]. It has been previously shown and our present proteomic data corroborates this that MDSCs are highly dependent on mitochondrial metabolism.

Another striking difference between TAMs and m-MDSCs was the differential expression of HIF1-alpha in the latter. This information agrees with results of Corzo and colleagues [10] on the role of HIF1-alpha in MDSC differentiation. However, it is important to remark that myeloid subsets differentiated in our experiments were generated in normoxic conditions, but they produce instead large amounts of NO due to increased iNOS expression [5,27]. It has been previously published by Li and colleagues [38] that high levels of NO and iNOS are stabilize HIF1-alpha by nitrosylation of cysteine 533. This suggests a role of NO in stability of HIF1-alpha in our ex vivo MDSC differentiation system. Strong up-regulation of lysosomal targets in TAMs included different cathepsins. These results corroborated previously published data of Yang and colleagues [39] who showed macrophage polarization towards M2, and their role in promotion of tumor progression by up-regulation of cathepsin S-driven autophagy. The increased activity of cathepsins was previously associated to the TAM phenotype of macrophages. Their activity is also associated to toll-like receptor cascade activation [40,41,42], identified as another TAM-associated reactome in our study. In contrast to TAMs, m-MDSCs are characterized by a strongly up-regulated metabolism of nitrogen compounds. Numerous studies in the literature identify high iNOS activities in m-MDSCs associated to T cell suppression [43].

Different studies have described the immunosuppressive characteristics of TAMs and m-MDSCs. Side-to-side comparisons between TAMs and m-MDSCs revealed key differences. While TAMs were characterized by high expression of MHCII, m-MDSCs were deficient in it. Interestingly, TAMs overexpressed the C1q component of the complement cascade, while m-MDSCs upregulated the expression of the C3 factor instead. C1q overexpression by macrophages was recently associated with a strongly immunosuppressive microenvironment and up-regulation of the immune checkpoints PD-1, LAG3, PD-L1 and PD-L2 [44]. It has been also previously shown by Ching and colleagues that C3 is responsible for polarization and differentiation of MDSCs [45]. Additionally, TAMs were characterized by BTN1A1 upregulation, a molecule identified as an inhibitor of T cell proliferation [46]. Finally, we identified ILF2 upregulation in TAMs. Recently, the PRMT6-ILF2 axis has been shown to be responsible for tumor progression and induction of pro-tumorigenic cytokine expression (MIF, IL8) [47]. Additionally, m-MDSC upregulated the expression of matrix metaloproteases MMP8 and MMP9, and proinflammatory molecules S100a8 and S100a9. The role of these factors in the immunosuppressive activity of MDSCs has also been studied before [48,49].

A key question in MDSC biology is to what extent monocytic MDSCs differ from granulocytic MDSCs. To our knowledge, this is the first time that a high-resolution proteomic comparison has been performed in highly-purified monocytic and granulocytic MDSCs. This technique has advantage over others such as single-cell sequencing as the latter considers a very limited number of expression targets in a sample of very few cells. In our analyses, one of the major differences is that granulocytic MDSCs has altered process regulating gene expression, reorganization of DNA structure and alternative RNA splicing. In previous studies, alteration of these processes correlate with an immediate inflammatory response program in neutrophils [50]. Moreover, we observed activated pathways regulated by FGF signaling. Monocytic-MDSCs were compared to granulocytic subsets, leading to the identification of several differentially expressed targets, as well as transcription factors associated to their differential proteomes. When comparing the predicted differential transcription factors, g-MDSCs showed transcription factor profiles associated to stronger pro-inflammatory responses, such as ATF-6, EP300 and SOX9. Moreover, while m-MDSCs increased SMAD7, the g-MDSC proteome was associated to SMAD3 instead. During recent years, the body of evidence on the immunosuppressive role of the granulocytic MDSC subset is growing. However, while there is a consensus that g-MDSCs are different from neutrophils from healthy donors, there is still continued confusion on their specific phenotype, as there is high variability among markers of polymorphonuclear subsets isolated form tumor-bearing mice and cancer patients. Recent work by Veglia and colleagues [51] characterised two subsets of polymorphonuclear granulocytic cells present in cancer patients and tumor bearing animals with immunosuppressive character. It has been previously published by us and others that granulocytic MDSC are matured monocytic MDSC [5,10,52].

Overall, our ex vivo-differentiation system can produce very large numbers (up to ten million cells per preparation) of myeloid cells that fully resemble immunosuppressive tumor-infiltrating subsets. All tested characteristics and phenotypes are in complete agreement with those from all published studies. Importantly, this differentiation system does not require the induction of tumors in mice, or direct isolation of tumor-infiltrating myeloid cells, a process that is technically challenging. Hence, these ex vivo-differentiated myeloid cells can be used as models to study tumor-induced immunosuppression, but also could be used as cellular therapies in experimental models of inflammatory diseases.

## Figures and Tables

**Figure 1 jpm-11-00542-f001:**
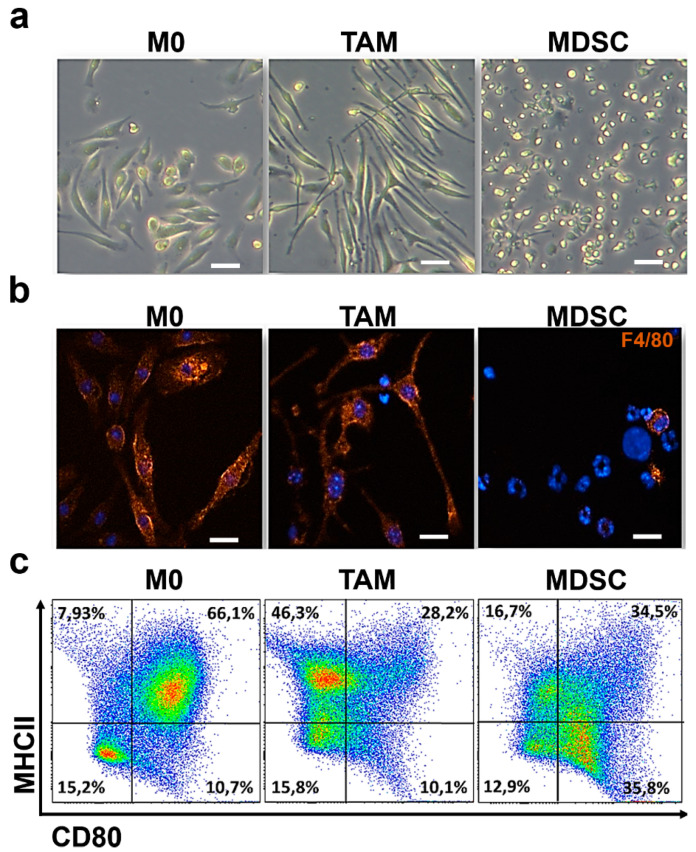
Phenotype of bone marrow-derived non-polarized macrophages, TAM-like macrophages and MDSC. (**a**) Phase contrast micrographs of cultures of ex vivo-differentiated uncommitted macrophages (M0), TAMs and MDSCs as indicated. Bars represent, 100 μm. (**b**) Fluorescence microscopy pictures of immunocytochemistry staining of the macrophage-specific marker F4/80 (orange) and nuclei stained with Hoesch (Blue). Macrophage and TAM preparations highly expressed F4/80, while MDSCs show typical multilobed or annular nuclei. Bars represent 100 μm. (**c**) Flow cytometry density plots of MHCII and CD80 expression in uncommitted macrophages (M0), TAMs and MDSCs as indicated. The gates were established according to unstained control cells. Percentage of events within each gate are indicated in the graphs.

**Figure 2 jpm-11-00542-f002:**
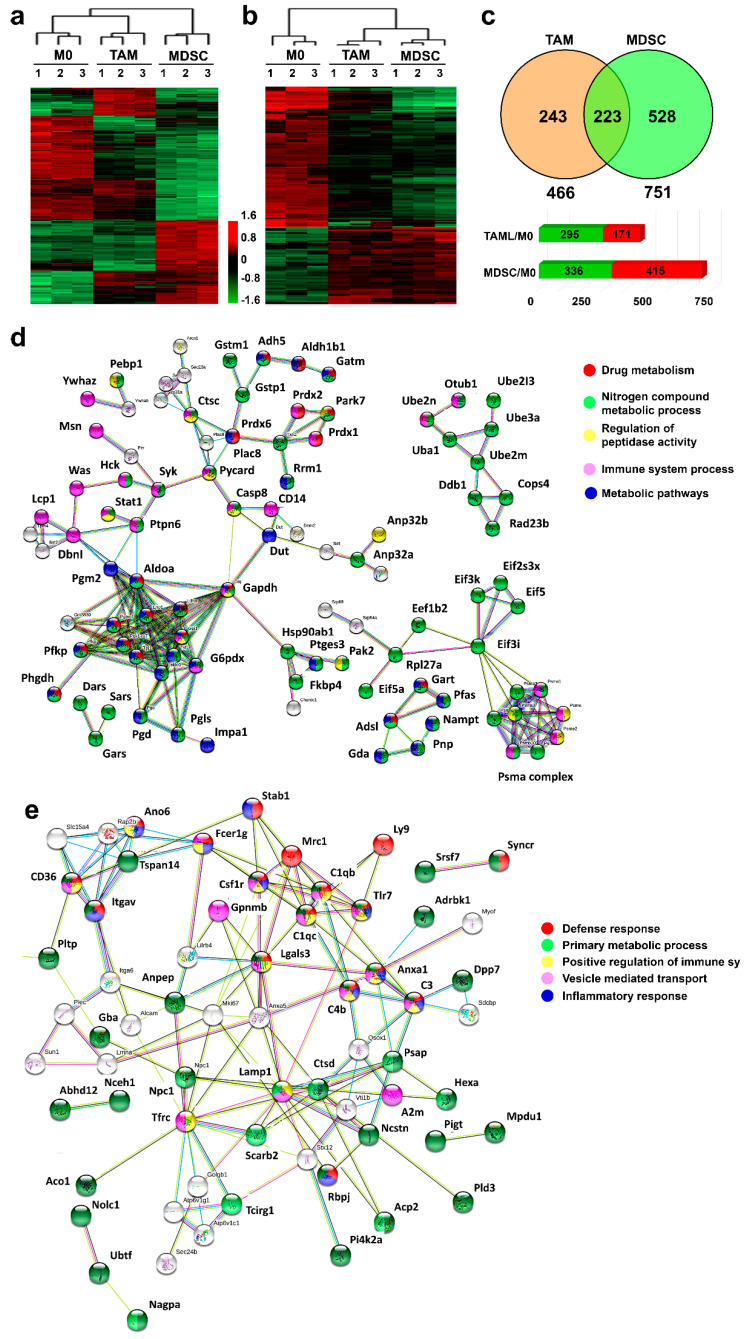
Lineage and cancer-specific interactomes in ex vivo-differentiated myeloid subsets. (**a**) Hierarchical unbiased clustering representing the differentially expressed proteins (ANOVA, *p* < 0.01) between M0, TAM and MDSC cell culture replicates (n = 3) as indicated by 1, 2 and 3. The heatmap shows differential protein expression modules associated to lineage. (**b**) as in (**a**) with differential protein expression modules regulated by cancer cell-polarization. Differentially-regulated proteins were classified using the proteome of uncommitted M0 macrophages as a normalizing control. Red and green, up and down-regulated proteins, respectively. The color-coded fold-change is represented next to the heatmaps as Log10. (**c**) The top Venn diagram represents differential proteins in TAMs and MDSCs identified in (**a**) and (**b**). The bar graphs show the number of differentially expressed proteins in MDSCs and TAMs compared to the M0 proteome. Upregulated proteins are indicated with red bars, and downregulated with green. (**d**) Shared upregulated functional interactomes in TAMs and MDSCs reconstructed with STRING. The color codes represent protein nodes belonging to the indicated pathways. (**e**) Shared downregulated functional interactomes in TAMs and MDSCs reconstructed with STRING. The color codes represent protein nodes belonging to the indicated pathways.

**Figure 3 jpm-11-00542-f003:**
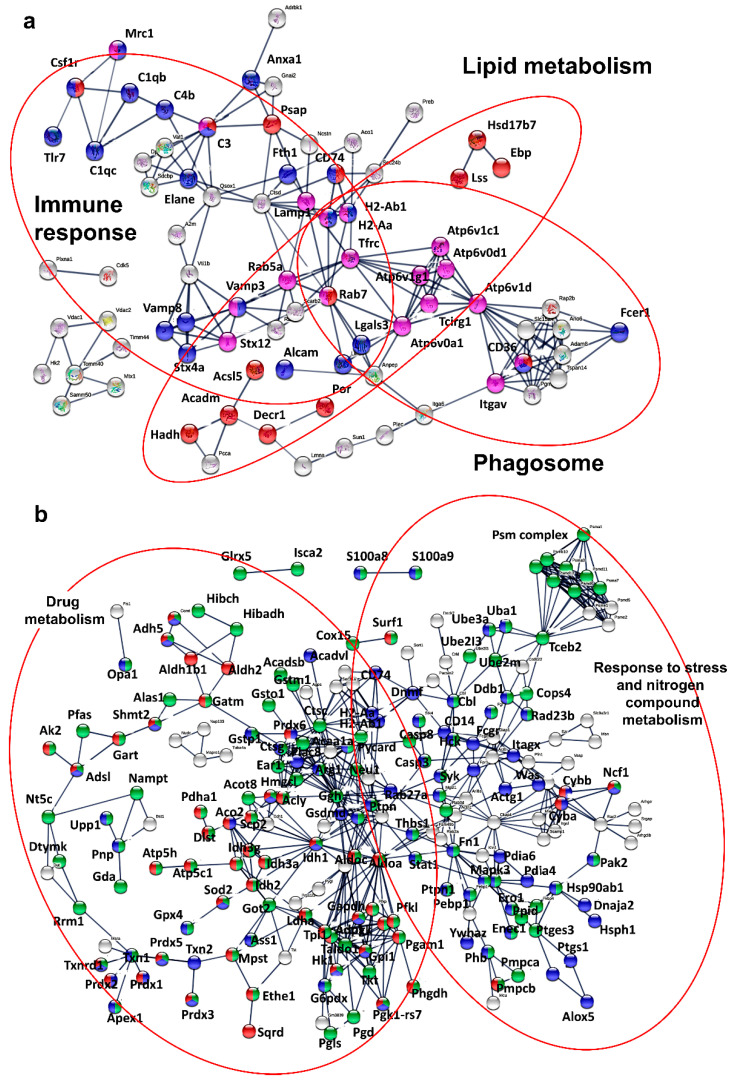
Cancer-specific interactomes in ex vivo-differentiated myeloid subsets. (**a**) TAM-specific upregulated functional interactomes compared to M0 macrophages reconstructed with STRING. The color codes represent protein nodes belonging to the indicated pathways. (**b**) Same as in (**a**), but with MDSC-specific upregulated functional interactomes.

**Figure 4 jpm-11-00542-f004:**
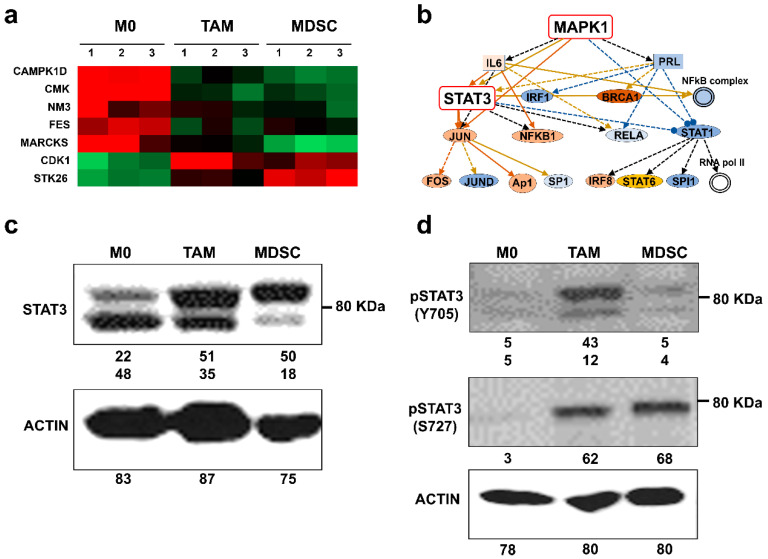
Differentially expressed kinases in ex vivo-differentiated myeloid cells. (**a**) Heatmap of hierarchical clustering representing the differentially expressed kinases identified by proteomics between M0 and tumor-polarized TAMs and MDSCs. (**b**) Functional interactome of kinase networks reconstructed with the Ingenuity Pathways Analysis Tool using the differentially-expressed proteomes in tumor-polarized myeloid cells. Direct interactions/processes are indicated with unbroken arrows. Indirect interactions/processes by dotted arrows. Blocked arrows indicate inhibitory interactions. (**c**) Western blot of STAT3 expression in ex vivo-differentiated myeloid cell types as indicated. STAT3 is resolved in two bands corresponding to distinct phosphorylation states. (**d**) Western blot analyses of phosphorylated STAT3 isoforms as indicated next to the blots, in ex vivo-differentiated myeloid cell types. Westerns were performed in the same membrane after stripping, and equal protein loading was ensured by quantifying protein concentration in lysates using Bradford. The numbers below the western blots indicate band intensities as measured with ImageJ.

**Figure 5 jpm-11-00542-f005:**
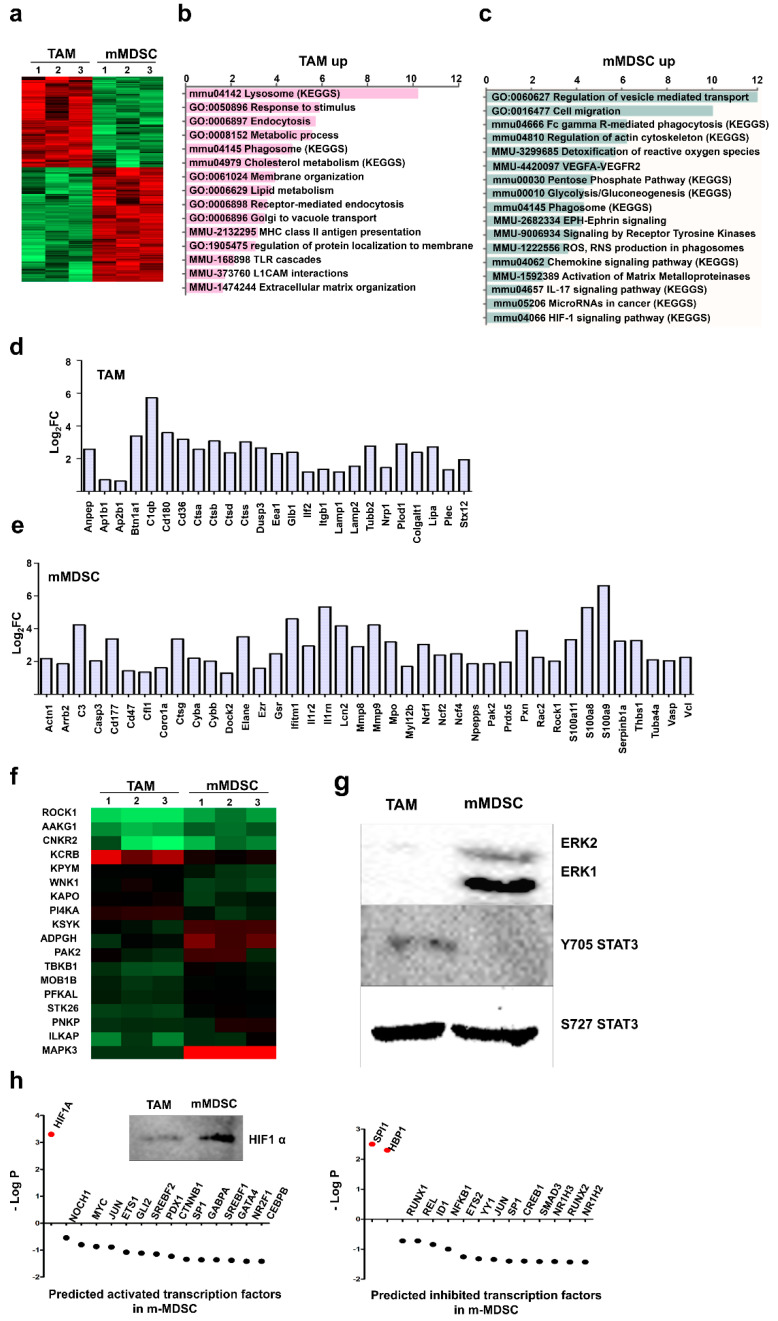
Differential proteomes between TAM and monocytic MDSCs discriminate both populations. (**a**) Heat map of the differentially expressed proteins (*p* < 0.01) between the indicated samples with 3 independent biological triplicates of TAM and m-MDSC cultures. Red and green, up and down-regulated proteins, respectively. (**b**) The bar graphs represent the enrichment of the differentially pathways in the TAM proteome, as analyzed by gene ontology and KEGGs pathways. (**c**) Same as (**b**) but using the differentially up-regulated proteome in monocytic MDSCs. (**d**) The bar graph represents the fold change in expression of the indicated selected targets in the differential proteome of TAMs which modulate immune responses in TAM. (**e**) Same as (**d**), but using the differential proteome of monocytic MDSCs. (**f**) Heat map representing the differential expression of the indicated kinases identified in the TAM and m-MDSC proteomes. Kinase expression in both subsets. (**g**) Western blots of the indicated kinases in TAMs and m-MDSCs as shown. (**h**) The dot plot graphs represent the probability of each indicated transcription factor to be either activated (left graph) or inhibited (right graph) in monocytic MDSCs compared to TAMs using the Tfacts algorithm. Transcription factors with statistical significance of association to the differential m-MDSC proteome according to probability (*p* < 0.05) and false discovery rates (*p* < 0.05) are highlighted in red. The western blot shows HIF1-alpha expression in TAMs or monocytic MDSCs as indicated.

**Figure 6 jpm-11-00542-f006:**
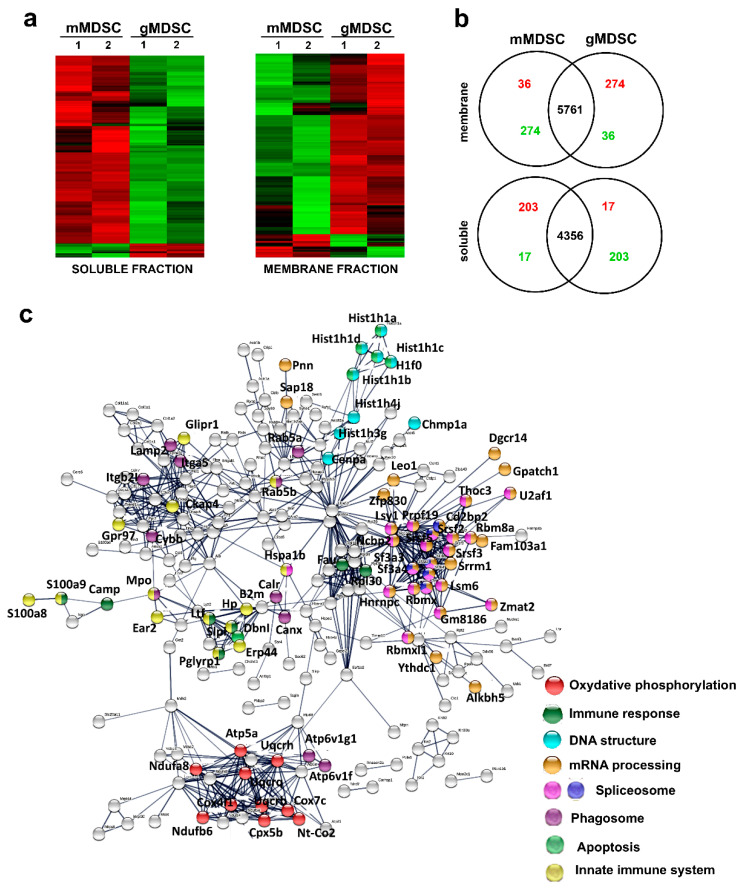
Monocytic and granulocytic-MDSCs differential pathways. (**a**) Heat maps of differential protein expression profiles (FC < 2Std) between monocytic and granulocytic MDSC cell cultures (duplicate independent biological replicates and purifications) as indicated, within the soluble fraction (left) and membrane fraction (right). Red and green, up and down-regulated proteins, respectively. (**b**) Venn diagrams of differentially upregulated (red) and down-modulated (green) proteins in membrane and soluble fractions between m-MDSCs and g-MDSCs as indicated. (**c**) g-MDSC-specific upregulated functional interactomes compared to m-MDSC cells reconstructed with STRING. The color codes represent selected protein nodes belonging to the indicated pathways.

**Figure 7 jpm-11-00542-f007:**
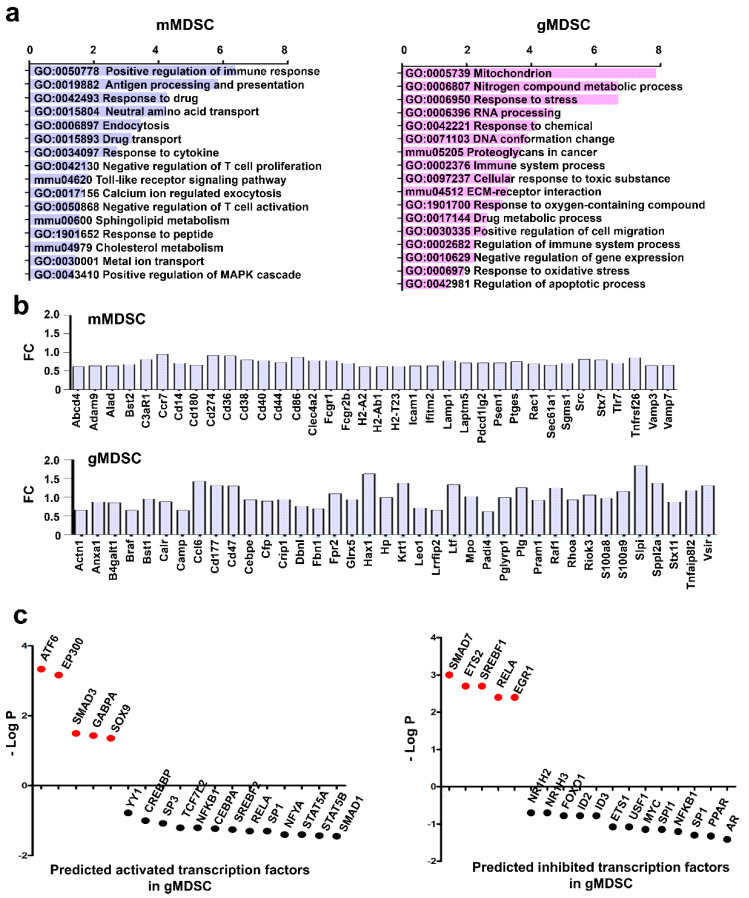
Monocytic and granulocytic-MDSCs differential pathways. (**a**) The bar graphs represent the enrichment of the differentially up-regulated pathways in the g-MDSC or m-MDSC proteome as indicated, as analyzed by gene ontology and KEGGs pathways. (**b**) The bar graphs represent the fold change in expression of the indicated selected immune modulators in the differential proteome of g-MDSC or m-MDSC as indicated. (**c**) The dot plot graphs represent the probability of each indicated transcription factor to be either activated (left graph) or inhibited (right graph) in g-MDSCs compared to m-MDSCs using the Tfacts algorithm. Transcription factors with statistical significance of association to the differential m-MDSC proteome according to probability (*p* < 0.05) and false discovery rates (FDR < 0.05) are highlighted in red.

**Figure 8 jpm-11-00542-f008:**
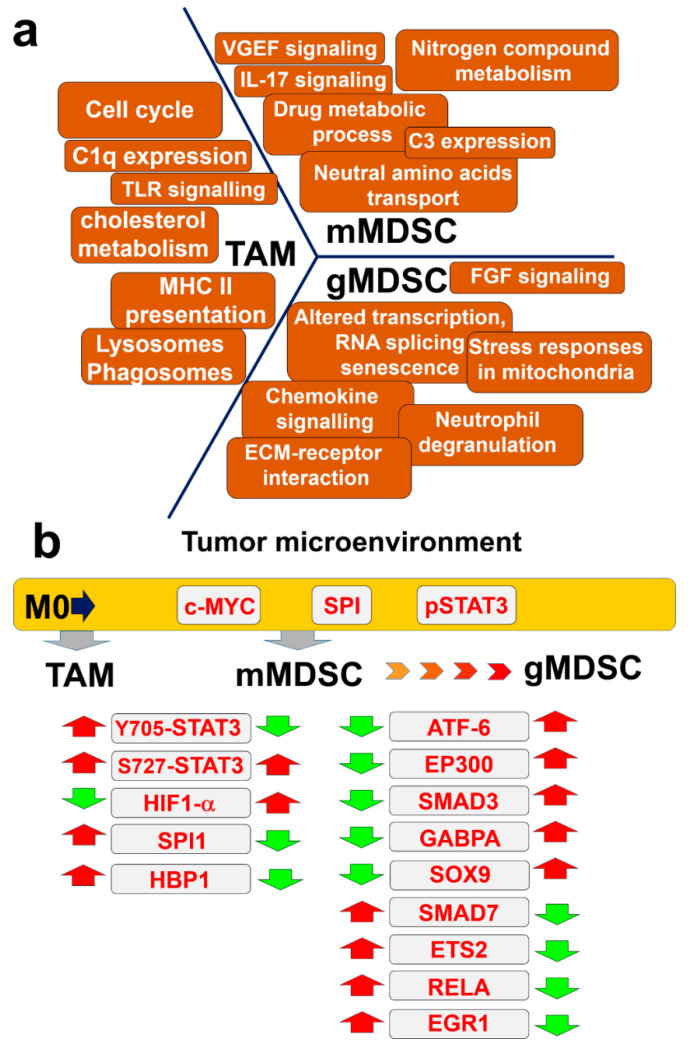
Proteome atlas of tumor-polarized myeloid cell subsets. (**a**) Processes associated to each specific myeloid subset, as indicated. (**b**) Hierarchy of transcription factors associated to each myeloid cell subset. Red arrows, up-regulated; green arrows, down-modulated.

## Data Availability

MS data and search results files were deposited in the Proteome Xchange Consortium via the JPOST partner with the identifier PXD025708 for ProteomeXchange and JPST001146 for jPOST (for reviewers: https://repository.jpostdb.org/preview/143364250608a8e370fba3; Access date 29 April 2021; Access key: 4446).

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
