# Peer review of "A Proteomic Atlas of Lineage and Cancer-Polarized Expression Modules in Myeloid Cells Modeling Immunosuppressive Tumor-Infiltrating Subsets"

_jpm, 2021, doi:10.3390/jpm11060542_

Round 1

Reviewer 1 Report

In this article, Blanco and his colleagues adapted their previously established ex vivo culture system to explore the differences between TAMs and other MDSCs. In the research, they conducted several proteomic experiments between different cell populations, such as MDSCs vs TAMs, m-MDSCs vs TAMs, as well as m-MDSCs vs g-MDSCs. Their results were comparable to those in references. Further, some of the goals mentioned in the introduction seems to have not been proven in the results, such as the ontogenic relationships between those myeloid immunosuppressive populations. In addition, as mentioned in this article, MDSCs consist of two distinct populations, m-MDSC vs g-MDSC. The authors should provide the percentage of these two populations for further interpretation. In addition, the following items should be clarified in the revised manuscript, if any.

  1. The resolutions of several figures (such as Fig 2d-g, and 3b) are poor, and revision of these figures are suggested.
  2. Internal control should be added in the Fig 4g, and the quality of this image should be improved as well.
  3. The labels on Fig3c and 3d were not clear enough to readers. The molecular sizes and the intensities of indicated bands should be presented in these figures.
  4. The constructed functional protein networks stated in line 456 should be provided in this article or in supplementary materials.
  5. The figures related to the statement in lines 457-463, read as ”Major differences were found on the profiles of proteins involved in immune response modulation and metabolic interactomes, although all of them related to negative regulation of T cell and interferon responses.” should be provided in this article or in supplementary materials.
  6. There is no statement of Fig5d in the manuscript.
  7. The “Discussion” section seems not finished.

Author Response

Reviewer 1.

Overall, the Reviewer suggests some major changes and clarifications that we have addressed point by point as follows:

  1. Reviewer 1 states that the results obtained in this study were comparable to those in references.

Indeed, while our ex vivo differentiation system has been set up by our group, the results are equivalent to the expected characteristics for immunosuppressive myeloid cells as described in the literature. Therefore, our system can be used to expand very large numbers of immunosuppressive cells with characteristics identical to tumor-infiltrating subsets. These cells can then be used as experimental models, and also as potential cellular treatments for inflammatory disorders. Hence, this statement is not a weakness, but a strength for our study. To clarify this point, we have completed the discussion stressing this fact as follows:

  • Last paragraph of the discussion: “Overall, our ex vivo-differentiation system can produce very large numbers (up to ten million cells per preparation) of myeloid cells that fully resemble immunosuppressive tumor-infiltrating subsets. All tested characteristics and phenotypes are in complete agreement with those from all published studies. Importantly, this differentiation system does not require the induction of tumors in mice, or direct isolation of tumor-infiltrating myeloid cells, a process that is technically challenging. Hence, these ex vivo-differentiated myeloid cells can be used as models to study tumor-induced immunosuppression, but also could be used as cellular therapies in experimental models of inflammatory diseases.”
  1. 2. Reviewer 1 is of the opinion that some of the goals mentioned in the introduction seem to have not been proven in the results, such as the ontogenic relationships between the myeloid immunosuppressive populations.

Although it is true that studies directed to prove ontogenic relationships have not been performed, cluster analyses highlight these relationships according to proteomic similarities. Therefore, the statement by the Reviewer is not entirely accurate. Nevertheless, Reviewer 1 is right stressing this point. Therefore, we have toned down our initial goals exposed in the introduction as follows:

  • Abstract, we have modified a sentence as follows: “Full functional interactome maps have been generated to characterize at high resolution the relationships between the three main myeloid tumor-infiltrating cell types”
  • Last paragraph of the introduction: “These analyses shed light on the similarities and differences between the myeloid immunosuppressive populations, their potential mechanisms of immune regulation, and highlighted differences identifying each cell type with unique proteomic fingerprints. “
  1. Reviewer 1 suggests to provide the percentage of m-MDSCs and g-MDSCs in the appropriate parts of the manuscript to help interpretation.

We agree with this point and have added the following:   

  • In results, page 8, we have added the following: “Then, we wondered whether we could identify cancer-regulated pathways that were specific for either TAMs or MDSCs. Therefore, the proteome of M0 uncommitted macrophages was used as a normalizing control between the proteomes of TAM and MDSC preparations, uncovering 466 and 751 differentially expressed proteins in TAMs and MDSCs, respectively (Figure 2c). MDSCs preparations in these experiments consisted of 40% monocytic and 60% granulocytic subsets. These proteins were then separated into overexpressed or down-modulated proteins. Then, the differential functional interactomes up-regulated by TAMs or MDSCs were constructed with STRING.”
  1. The resolution of several figures is poor.

We have revised the resolution and the figures have been re-made to improve them. In the process, we had to break the original figure 2 in two figures, Figure 2 and Figure 3. And the addition of an additional Figure 6. Due to the high resolution of the pictures, it was the only way to introduce them in the text. This has resulted in a significant improvement in our manuscript.

  1. The Reviewer suggested that an internal control should be added in Figure 4g and the quality of this image should be improved as well.

We apologise for not being sufficiently clear. We found in our experiments that phosphorylated S727-STAT3 did not vary in our myeloid cells, and that is the reason that from that moment on we decided not to use actin as a loading control in these experiments. In fact, we found in the proteomics data that granulocytic and monocytic MDSCs had different actin expression levels. To make it clear to the readers, we added the following sentence in page 15: “In this case, we used S727-phosphorylated STAT3 as a loading control, as we found in this study not to be different between our ex vivo-differentiated myeloid cells.

We have also tried to improve the quality of the image by removing background in two of the western blots, although we would have preferred the westerns to be as unmanipulated as possible.

  1. The Reviewer comments that the labels of Fig 3c and 3d were not clear enough. That the molecular sizes and intensities of the indicated bands should be presented in this figure.

To comply with the Reviewer´s request, we have increased the letter sizes for these images. The molecular size of the western blot close to STAT3 has been added and the band intensities provided using Image J. We have therefore modified the figure legend as follows (Now figure 4): “Figure 4. Differentially expressed kinases in ex vivo-differentiated myeloid cells. (a) Heatmap of hierarchical clustering representing the differentially expressed kinases identified by proteomics between M0 and tumor-polarized TAMs and MDSCs. (b) Functional interactome of kinase networks reconstructed with Ingenuity Pathways analysis using the differentially-expressed proteomes in tumor-polarized myeloid cells. Direct interactions/processes are indicated with unbroken arrows. Indirect interactions/processes by dotted arrows. Blocked arrows indicate inhibitory interactions. (c) Western blot of STAT3 expression in the ex vivo-differentiated myeloid cell types as indicated. STAT3 is resolved in two bands corresponding to distinct phosphorylation states. (d) Western blot analyses of phosphorylated STAT3 isoforms as indicated next to the blots, in the ex vivo-differentiated myeloid cell types. Westerns were performed in the same membrane after stripping, and equal protein loading was ensured by quantifying protein concentration in lysates using Bradford. The numbers below the western blots indicate band intensities as measured with ImageJ.“

  1. The constructed functional protein networks stated in line 456 should be provided in this article or in supplementary materials.

The Reviewer is right. We have added the protein networks. This has resulted in Figure 6 to be split into Figure 6 and 7. The corresponding figure legends have been amended to comply with the request.

  1. The Reviewer comments that “The figures related to the statement in lines 457-463, read as ”Major differences were found on the profiles of proteins involved in immune response modulation and metabolic interactomes, although all of them related to negative regulation of T cell and interferon responses.” should be provided in this article or in supplementary materials. In addition, in the next point the reviewer comments that “There is no statement of Fig5d in the manuscript.”.

We apologise to the Reviewer. The figures related to the statement in lines 457-463 corresponds to the statement of Fig5d, which was lost during the preparation of the manuscript. Hence, we have added the corresponding call to Fig5d (now Fig 7) as follows:

  • Page 15: “Major differences were found on the profiles of proteins involved in immune response modulation and metabolic interactomes, although all of them related to negative regulation of T cell and interferon responses (Figure 7b).
  1. The “Discussion” section seems not finished.

Indeed, there was an unintentional abrupt end to the Discussion section, which we have completed as described in the first point of our response to the Reviewer.

Reviewer 2 Report

The authors presented here a very interesting paper describing the differentiation, identification and characterization of different myeloid population using various biochemical and bioinformational methods. The authors successfully differentiated M0 cells to MDSCs and TAMs resembling tumor subsets, and identify the different tumor-inducing functions, kinases profiles and transcription factors. The results supported the conclusion.

There are some minor problems here:

  1. No reference in Paragraph 1 of introduction, please add proper references here.
  2. Please remake Figure 2, the figure legend covered part of your figure.
  3. Figure 3B, resolution is low, if possible could you make it better?

Author Response

Reviewer 2.

We thank the reviewer for the positive comments. We have addressed the minor problems highlighted by the Reviewer as follows, point by point (and highlighted in the manuscript with yellow):

  1. No reference in Paragraph 1 of introduction, please add proper references here.

Thanks for pointing out to us that there are no proper references in the first paragraph. We have added the appropriate references, and highlighted them in yellow.

  1. Please remake Figure 2, the figure legend covered part of your figure.

We appreciate the comment from the Reviewer. We have introduced the color-coded bar in the figure, and completed the figure legend to fully match the figure. The changes are highlighted in yellow.

  1. Poor resolution of Figure 3.

We have substantially improved the resolution of Figure 3B. We have also completed the figure legend as follows (“Direct interactions/processes are indicated with unbroken arrows. Indirect interactions/processes by dotted arrows. Blocked arrows indicate inhibitory interactions.”).

Round 2

Reviewer 1 Report

In the previous review, I focused on two main issues. The first is the quality of some images and figures, which affected the interpretation of a scientific article a lot. The authors have provided high-resolution figures, and these results are more easily readable.

My second main question is about some missing information in the article. The authors did provide sufficient information in this revised version. In addition, the differences between each cell population of inhibitory myeloid cells are highlighted and thus more convincing to me.